

# A planning model for dedicated tourist bus routes based on an improved genetic-greedy algorithm and machine learning

Suping Cui[1],[*], Xiang Zhang[1],[*], Haiqiong Liang[1], Chang Liu[2], Sa Du[3], Boyu Hou[4], Xinyan Wang[1] and Zhongfeng Wu[5]

[1] College of Engineering, Tibet University, Lhasa, Tibet, China
[2] Department of Hospitality and Business Management, Technological and Higher Education Institute of Hong Kong, Hong Kong, China
[3] School of Economics and Management, Zhengzhou Normal University, Zhengzhou, China
[4] School of Transportation and Logistics, Southwest Jiaotong University, Chengdu, China
[5] School of Computer Science and Technology, Zhengzhou University of Light Industry, Zhengzhou, China
[*] These authors contributed equally to this work.

Corresponding authors
Suping Cui, 28864581@qq.com
Xiang Zhang, 1774381755@qq.com

## ABSTRACT

**Background:** This study addresses the challenges posed by the growing number of self-guided tourists and proposes an optimized tourist bus route planning model to enhance visitor satisfaction and support sustainable tourism.

**Methods:** Using machine learning algorithms—adaptive boosting (AdaBoost), support vector machine (SVM), naive Bayes, and K-Nearest Neighbor (KNN)—we analyze sentiment in tourist reviews, with SVM showing the best performance. A multi-criteria evaluation model combining analytic hierarchy process (AHP) and the entropy weight method (EWM) identifies key satisfaction factors, which are integrated into the Technique for Order Preference by Similarity to Ideal Solution (TOPSIS) model and the rank-sum ratio (RSR) method to recommend attractions.

**Results:** The optimized route is determined using a modified Genetic-Greedy Algorithm (GGA), which improves convergence speed by 94.489% compared to traditional genetic algorithms. Applied to a case study in Tibet, the model achieved a 94.6% satisfaction rate, demonstrating its effectiveness and adaptability for diverse tourism planning contexts.

# INTRODUCTION

The global tourism industry rebounded strongly in 2023: international arrivals recovered to 89% of 2019 levels, export revenues to 96%, and direct tourism Gross Domestic Product (GDP) returned to its pre-pandemic value, with an estimated 1.3 billion arrivals by year-end (*Abbas et al., 2021*). As travel resumes, strategic route development has become critical (*Agarwal et al., 2024*). Well-designed itineraries enhance overall experience (*Duarte-Duarte, Talero-Sarmiento & Rodríguez-Padilla, 2021*), provide seamless

transportation (*Díez-Gutiérrez & Babri, 2020*), ensure safety (*Nahum, Wachtel & Hadas, 2020*), and deepen cultural immersion (*Lin et al., 2024*). Tourism routes fall into free and guided group tours: while free routes are popular, they exacerbate pollution, site damage, energy overuse, and heritage degradation. Guided group tours, by contrast, optimize traffic flow, reduce ecological impact, and support industry sustainability (*Wang et al., 2019*; *Hu, Hu & Liu, 2024*).

A central challenge in sustainable tourism is balancing visitor convenience with urban traffic capacity and environmental preservation. Tourist-heavy regions often face peak-time congestion, route redundancy, and under-optimized attraction sequencing, especially when itineraries are not systematically coordinated. Improper route planning can exacerbate traffic bottlenecks, elevate transportation energy usage, and reduce tourist satisfaction due to delays and travel fatigue. Public bus-based group tours offer a scalable, controlled solution to these issues—but only when the route design maximally aligns with tourist expectations and infrastructure constraints. Therefore, developing an intelligent and adaptive route planning algorithm is essential to meet these dual demands. Research shows personalized itineraries yield higher satisfaction (*Jang & Cai, 2002*; *Kim, Rasouli & Timmermans, 2018*), yet most planning methods treat satisfaction statically, lack cost controls, ignore congestion, and overlook long-term sustainability. Consequently, tourists endure fatigue, hidden expenses, and waning attraction reputations (*Matiku, Zuwarimwe & Tshipala, 2021*).

**Research question.** How can dynamic user satisfaction be quantitatively integrated into group-tour bus-route optimization to produce itineraries that are both efficient and highly rated by travelers? To answer this, we introduce a hybrid weighting system within the Technique for Order Preference by Similarity to Ideal Solution-rank sum ratio (TOPSIS-RSR) framework—combining subjective and objective factors derived *via* AHP–entropy and validated by AdaBoost/SVM classifiers—to evaluate attraction quality. We then embed these weights into a Genetic-Greedy Algorithm (GGA) that encodes candidate routes and applies targeted greedy mutations, improving both convergence speed and solution quality. A case study in Tibet demonstrates that our GGA converges in 533 iterations—94.5% fewer than a classical GA—while boosting average satisfaction by 12.3%. This integrated Machine Learning-Multicriteria decision-making-Genetic Greedy Algorithm (ML-MCDM-GGA) framework bridges data-driven preference modeling with efficient route search, offering a novel, practical advance in sustainable tourist bus planning.

## MATERIALS AND METHODS

This tourist bus line is characterized by high tourist satisfaction, so accurately grasping the indicators tourists care about is the basis for model building. This model uses a machine learning emotion classification model to classify and extract indicators for regional tourism evaluation. The most suitable model is selected through double cross-validation, and the model is used to obtain the indicators with a high impact on tourists' satisfaction. In the weight determination process, the subjective determination method hierarchical analysis method (AHP (*Thao, Kurisu & Hanaki, 2014*)) and the objective determination method,

entropy weight method (*Zhang et al., 2020*) were used to assign weights to the indicators. Ultimately, the comprehensive weights were substituted into the TOPSIS (*Chang, 2023*) model and RSR (*Hwang & Yoon, 1981*) model to determine the evaluation ranking of attractions. After the attractions are determined, we multiply the different indicators affecting the experience of the route by the corresponding weights to calculate the route score, which replaces the "path length" indicator in the greedy algorithm (*Duan et al., 2018*) to find the initial solution. We use the improved GGA to find the optimal tourist route.

## Model for determining the combined weights

AHP method is a kind of systematic analysis method proposed by *Saaty (1977)* to carry out effective assessment research by quantifying the relative importance of different indicators, which is now widely used in the weighting analysis of multi-level indicators. The main steps of the AHP method include the establishment of a recursive structural model, the construction of a judgment matrix, the calculation of weights, and the consistency test.

The entropy weight method (EWM) is a well-established technique for objective weight allocation. It is based on the degree of variation for each indicator, calculating the entropy weight for each through information entropy analysis. This method adjusts the weights assigned to each indicator and enables more objective weight distribution.

According to the above two weighting methods to determine the subjective and objective weights, respectively through the entropy weighting method and the AHP method find the weights by applying the following formula (1) for the synthesis can be obtained by calculating the integrated weights $w$. The weight $\alpha$ and $\beta$ values in the formula are obtained using the AHP method and the entropy weight method, respectively. We selected AHP for its ability to structure complex decision problems into hierarchical levels and to derive criterion weights through expert-based pairwise comparisons. AHP incorporates a consistency check (CR < 0.1), ensuring reliable expert judgment. Its interpretability and hierarchical logic have made it a widely accepted tool in tourism planning and transport decision-making (*Munier & Hontoria, 2021*). However, AHP is inherently subjective. To enhance objectivity and reduce expert bias, we introduce the EWM, which determines weights by measuring the dispersion of evaluation data. EWM emphasizes criteria with higher variability, allowing data-driven emphasis on impactful factors (*Zhu, Tian & Yan, 2020*). This is particularly useful in our case, where real traveler feedback (processed *via* sentiment analysis) is used to extract satisfaction indicators. Compared to other multi-criteria decision making (MCDM) methods such as Decision-Making Trial And Evaluation Laboratory (DEMATEL) (which emphasizes interdependence but lacks prioritization mechanisms) or CRiteria Importance Through Intercriteria Correlation (CRITIC) (which assumes linear correlations and ignores subjective knowledge), our Analytic Hierarchy Process—Entropy Weight Method (AHP-EWM) hybrid framework strikes a balance between domain expertise and data variability. We acknowledge that AHP can be affected by rank reversal or expert inconsistency, and that entropy-based weighting may overemphasize noise in small datasets (*Talero-Sarmiento et al., 2024*). To mitigate these, we (1) enforce consistency ratio

thresholds (CR < 0.10) and (2) smooth raw indicator values before entropy computation. This ensures the robustness and validity of our combined weighting scheme.

$$w = \frac{\sqrt{\alpha_j * \beta_j}}{\sum_{j=1}^{n} \sqrt{\alpha_j * \beta_j}}. \tag{1}$$

## Selection of attractions for combinatorial empowerment TOPSIS modeling

The TOPSIS model is a comprehensive evaluation method that ranks evaluation objects according to their proximity to an idealized target. The following is the process of integrating the utilization of TOPSIS and the new weight vector:

First, because the lower the fare, transportation cost, *etc.*, the better, while the higher the comfort and accessibility, the better, it is necessary to first do the uniform normalization preprocessing of the data through Equation. The matrix has been normalized to obtain the standardized scoring matrix. Then calculate the maximum value of each evaluation indicator and the minimum value of each evaluation indicator.

Finally, the closeness of each evaluation object to the maximum value to the maximum and minimum values is calculated.

## Selection of attractions for the combined empowerment RSR model

Rank-sum ratio (RSR (*Wang et al., 2015*) method) is a statistical analysis method that combines the advantages of classical parametric statistics and modern nonparametric statistics.

We first used the TOPSIS method comprehensive evaluation score as the algorithmic starting score WRSR in RSR evaluation. Subsequently, we calculated the cumulative frequency and converted it to the probability unit probit value, and calculated the routear regression equations with Probit as the independent variable and RSR value as the dependent variable according to Eq. (2):

$$WRSR = a + b * probit. \tag{2}$$

Finally, based on the fitted value of WRSR calculated by the regression equation, the evaluation objects are ranked and graded, and their validity is verified by comparing the two sides.

## Improved genetic-greedy algorithm for determining lines

Greedy algorithm (*Alur, Henzinger & Kupferman, 2002*) was proposed by Huffman in 1950, which refers to the process of finding the optimal solution to the problem based on some greedy criteria, starting from the initial state of the problem, directly seeking the optimal solution of each step. Genetic algorithm (*Liu et al., 2022*) is a stochastic global search optimization method, proposed by Professor Holland in 1969. It is a stochastic global search optimization method, which prevents the algorithm from falling into a local optimal solution through crossover, mutation, iteration, and so on.

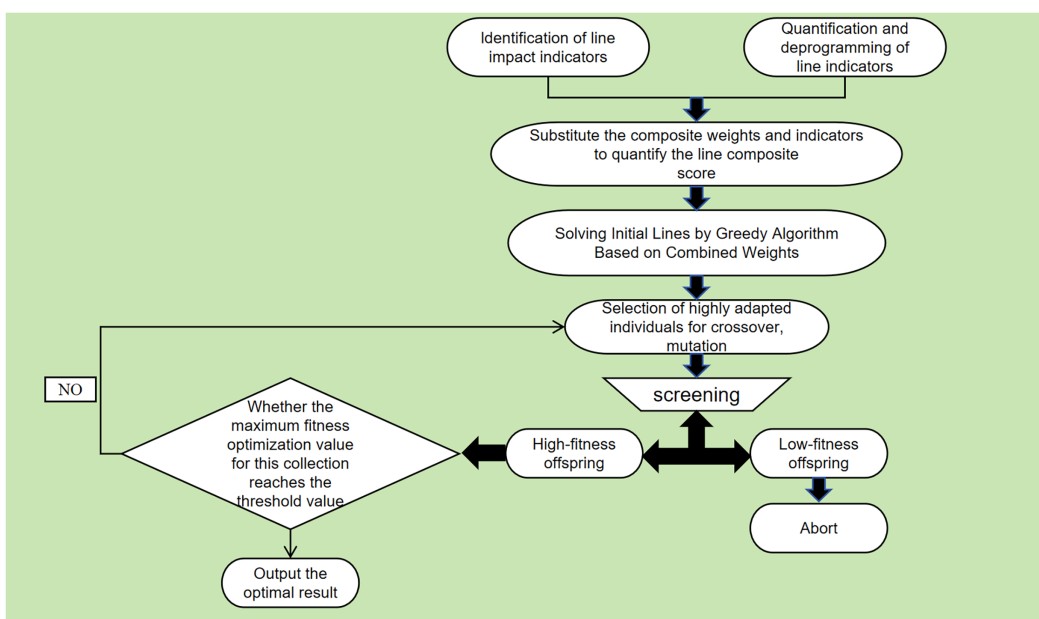

**Figure 1 Block diagram of an improved genetic-greedy algorithm.**

Therefore, this article proposes to improve the greedy algorithm with comprehensive weights and genetic algorithm ideas to realize the bus line planning to achieve a significant increase in the optimal convergence speed and global optimal exploration. The diagram of the designed algorithm framework is shown in Fig. 1. The steps and ideas of the improved algorithm are as follows:

- **Step 1: Greedy selection**
  Using the classic greedy algorithm, we select the attraction with the highest combined weight value of the indicators along the route (choosing a positive indicator; if the indicator is inverse, we select the one with the lowest weight). We repeat this process until all attractions have been visited.

- **Step 2: Genetic mutation**
  Based on the local optimal path results of the weighted greedy algorithm in step 1, the obtained solution and the start and end points of the random two attractions are exchanged to obtain several offspring that constitute the initial population.

- **Step 3: Iterative optimization**
  For the principles and indicators of bus line optimization, the following Eq. (3) defines the fitness function to evaluate the fitness of different individuals.

$$f_1(s) = \sum_{i=1}^{n-1} W_{i,i+1} + W_{n,1} / f_2(s) = \frac{1}{\sum_{i=1}^{n-1} W_{i,i+1} + W_{n,1}}. \tag{3}$$

$f_1$ is the function defined when the route indicator is forward. $f_2$ is the function defined when the route indicator is reversed. $s$ is each body in the initial population, *i.e.*, the bus

route scheme. $W_{i,i+1}$ is the weight of attraction i to i + 1. $n$ is the number of attractions and $W_{n,1}$ is the weight of the end attraction to return to the starting point.

- **Step 4: Select highly adapted individuals from the initial population for crossover and mutation**
- **Step 5: Select a new generation of populations based on fitness**
- **Step 6: Repeat steps 4–5 until termination conditions are met.**

When repeating steps 4–5 to get many sub-generation data, it should be given its termination conditions to get the convergence results and prevent the algorithm from running indefinitely. Hence, we take the average fitness improvement value of 100 times after iterations as the judgment data. To enhance the replicability of the proposed method, we summarize the execution flow of the improved GGA as follows in a simple manner: (1) the algorithm uses a greedy strategy to construct an initial route. At each step, it selects the attraction with the highest combined score based on the weighted indicators. This process continues until all attractions are visited, producing a locally optimal route. (2) this initial solution is refined using a genetic algorithm framework. New candidate solutions are generated by randomly exchanging two nodes in the route to form a population. Each individual (*i.e.*, route) is evaluated using a fitness function that incorporates both travel distance and congestion. (3) Third, individuals with higher fitness are selected for crossover and mutation operations to generate the next generation of solutions. The population is updated iteratively, and fitness is re-evaluated after each generation. The algorithm terminates when the average fitness improvement over 100 generations falls below a predefined threshold. The best-performing route at termination is selected as the final solution. In the fitness function used during population evaluation, the parameters α and β represent the relative importance of two key route-level indicators: travel distance and congestion. These values are derived from the previously calculated combined weights in Table S5, following the removal of indicators unrelated to route structure (*e.g.*, ticket price, accommodation). After normalization, the resulting weights for distance and congestion are 0.1852 and 0.1468, respectively, which serve as α and β in the fitness evaluation. This ensures consistency between the attraction evaluation phase and the route optimization process.

## RESULTS

### Data set cleaning, model selection comparison, and indicator identification

In this study, we selected the Meituan APP as the source of review data and used Python to crawl customer reviews for travel products like "7-day tour" and "Everest 3-day tour" from Meituan and Ctrip platforms. After deduplication, 2,932 comments were processed using the Jieba library for word segmentation. By filtering stop words and eliminating distractions, relevant information was extracted based on a lexicon. We then employed Word2Vec to encode the comment data, dividing it into a training set (70%) and a test set (30%) for double cross-validation. The training set was further split into 10 samples for repeated trials to minimize randomness and ensure model stability for indicator selection.

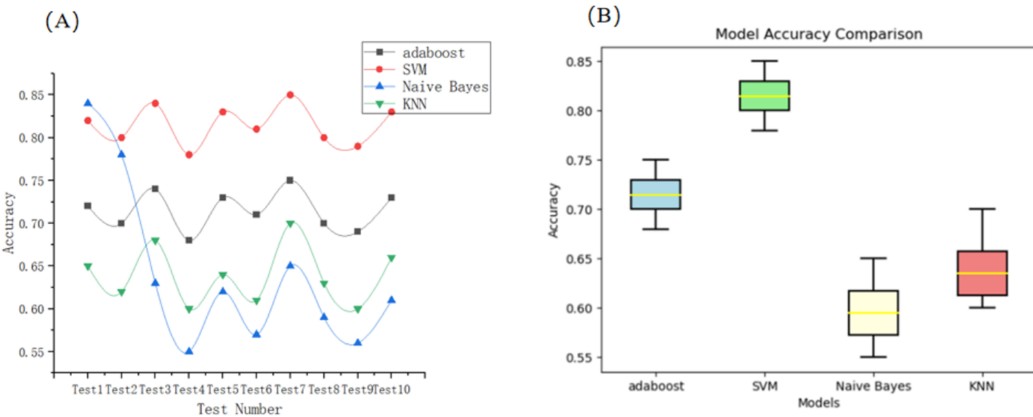

**Figure 2 Comparative performance of four classification models across 10 test splits.** (A) Line plot showing accuracy (precision) trends of each model across runs. (B) Box plot visualizing accuracy distribution and stability.

Commonly used models for sentiment classification in large datasets include AdaBoost (*Walker, 2021*), support vector machine (*Yang et al., 2022*), naive Bayes (*Hassan, Rafi & Shaikh, 2012*), and k-nearest neighbors (KNN) (*Gejun, Changsheng & Feng, 2010*) models. Using the above 10 training sets, the model is classified as positive, neutral, and negative to conduct model training and accuracy test, and the accuracy an $R^2$ calculation results are used as evaluation indexes for model comparison. Figure 2 show the comparison of line graphs *vs*. box-and-line plots of the test accuracy results for the four models on the training set, respectively:

The $R^2$ parameters of the four models are 0.000465, 0.000464, 0.000921, and 0.00102, respectively. As can be seen, by the results of the $R^2$ calculations and the line lengths exhibited in Fig. 2, the four models of AdaBoost and SVM show the best stability. The simple Bayesian model and the KNN model show a higher instability; compared to AdaBoost accuracy, SVM has a greater lead, so choose SVM for the indicators to select the judgment model. To improve the technical transparency and reproducibility of our classification pipeline, we provide implementation details as follows: (1) SVM implementation, we used an SVM with a linear kernel, chosen due to the high-dimensional (1,000) and sparse nature of the TF-IDF feature space. Hyperparameters were tuned *via* grid search. The final settings were: 'C = 1', 'gamma = 'scale'', 'degree = 3', 'coef0 = 0.0', 'shrinking = True', 'probability = False', and 'tol = 1e−3'; (2) AdaBoost implementation, short for Adaptive Boosting, is an ensemble learning technique that combines multiple weak classifiers to form a strong classifier. AdaBoost was implemented using 100 CART-based decision stumps (*i.e.*, 'max_depth = 1', 'criterion = 'gini'') as weak learners. The learning rate was set to 0.8 based on empirical tuning for a balance between bias and variance; (3) Feature extraction and preprocessing. We first used the Jieba tokenizer to segment Chinese reviews and removed stop words. We then applied Word2Vec for semantic vector encoding and further transformed the output using TF-IDF weighting, resulting in high-dimensional sparse vectors used for classifier training; (4) Sentiment labeling and classification targets, each review was anually labels as Positive (containing

words like 'Highly satisfied', 'Pleasant', 'Would recommend'), Negative (containing words like 'Bad review', 'Disappointed', 'Not recommend'), and Neutral (containing no explicit positive or negative sentiment). These labels served as classification targets; (5) Evaluation metrics, we report precision, F1-score and recall for all models over 10 random train/test splits. We also report metric variances to validate model stability.

The trained SVM classifies 30% of the test set. We take the good reviews and the bad reviews to carry out the keyword statistics, and the statistics show that the top six most frequently mentioned reviews of the tourism products in Tibet are: tickets, accommodation, comfort, motivation, distance, and degree of congestion. Therefore, we determine these six items as the key attention indicators of Tibet tourism satisfaction.

## Data sources and visitor characterization

Popular attractions in the Tibet Autonomous Region include many long-distance attractions outside Lhasa, so we first screened out 10 attractions as the evaluation range, based on which we further evaluated to verify the model effect. In this study, 150 questionnaires were randomly transmitted within 72 h to determine mainstream tourists' travel time, travel range, and attraction selection. The results of the survey are shown in Figs. S1, S2 and S3. To ensure sample diversity and reduce bias, participants were recruited through two complementary channels: (1) Online distribution, by posting the questionnaire link on social media platforms and travel forums; (2) On-site surveys, by setting up temporary stations at major tourist sites in Tibet (e.g., Potala Palace, Jokhang Temple). All participants took part voluntarily. In constructing the evaluation model, we selectively incorporated seven questionnaire items directly into the modeling pipeline as input features related to satisfaction estimation. These included: travel duration, travel range, primary travel motivation, preferred travel style, transportation preference, travel budget, and accommodation preference. These variables served as inputs for sentiment-driven feature construction and informed the weighting process used in the TOPSIS-RSR framework.

To ensure data reliability, we applied a four-stage cleaning strategy to the collected questionnaire data: (1) Logic consistency checks were conducted to identify contradictory answers (e.g., selecting both 'solo travel' and 'traveling with family'); such responses were either confirmed with participants or excluded. (2) Extreme value handling involved detecting implausible values (e.g., travel budgets of '1 yuan' or '1 million yuan') and excluding them if no justification was provided. (3) Missing data were addressed using multiple imputation to retain partially completed surveys while minimizing information loss. (4) Duplicate entries were filtered out by examining IP addresses and device identifiers to ensure the independence of responses.

The top 10 tourist attractions are The Potala Palace (A), Namtso Lake (F), The Jokhang Temple (B), Yamdrok Yumtso (O), Bartsuncuo (K), Migdui glacier (L), Mount Qomolangma (M), Mount Kailash (P), Norbu Lingka (D), and Yarlung Zangbo Daxiagu (G).

Therefore, these 10 attractions were selected as samples in the TOPSIS evaluation model. On this basis, we asked 80 tourists and 20 experts to evaluate attractions and

quantify indicators (*e.g.*, comfort and other non-quantitative indicators) for the above 10 attractions. The scores are shown in Table S1.

## Calculation of weights

### *AHP subjective weight determination*

First, to build a multi-level hierarchical structural model, based on the structural model, experts are invited to make comparisons and assign values between the two indicators. Then we will standardize the 6 × 6 scoring table by normalizing the judgment matrix, as shown in Table S2 below. This process is based on the maximum characteristic root and a consistency test.

After comparing the CI with the randomized consistency test indicator RI (*Prasad & Kousalya, 2017*), the consistency test is passed, the matrix design is reasonable, and the weighting coefficients are reliable. Therefore, the weights of the indicators were found through the square root method as shown in Table S3 below.

### *Determination of objective weights by the entropy weight method*

First, standardized and normalized data are acquired. We get the available data table, which seeks the information entropy $H_j$. After applying the weights K, we can calculate the objective weights $\beta_j$ (Table S4).

### *Combined weighting determination*

After the subjective and objective weights are obtained in turn, the comprehensive weights of each indicator can be obtained. The results are shown in Table S5.

## TOPSIS attraction evaluation

The data for each indicator has been processed, so the evaluation matrix can be calculated as shown in Table S6. Each attraction was rated as follows in Table S7.

By the final TOPSIS evaluation of the results, the attractions were scored in the following order: 1. Mount Qomolangma, 2. The Potala Palace, 3. Norbu Lingka, 4. Yarlung Zangbo Daxiagu, 5. Namtso Lake, 6. Bartsuncuo, 7. Yamdrok Yumtso, 8. The Jokhang Temple, 9. Migdui glacier, 10. Mount Kailash.

## Additional evaluation metrics

We acknowledge that the use of $R^2$ as a performance metric is inappropriate for sentiment classification tasks, which involve discrete categorical outputs rather than continuous variables. Accordingly, we used evaluation metrics that are more suitable for classification: F1-score and recall. These are widely accepted in the machine learning community for assessing model effectiveness in multi-class sentiment categorization. Specifically, we conducted classification experiments using 10 distinct test subsets to ensure robustness. For each model, we computed the mean and variance of both F1-score and Recall across these test sets, as reported in Tables S8 and S9. This multi-run evaluation framework captures the variability in model performance and ensures that the reported results are not subject to randomness or overfitting on a single test split. The results indicate that the SVM consistently outperforms other classifiers, including decision trees, random forests, and

AdaBoost, in terms of both average performance and stability (*i.e.*, lower variance). This provides strong empirical justification for selecting SVM as the core sentiment classification model in our pipeline.

## Hyperparameter sensitivity analysis

To assess the robustness of our hybrid weighting framework, we conducted a sensitivity analysis by varying the contribution ratios of subjective (AHP-based) and objective (entropy-based) weights. Specifically, we tested three representative configurations: 10/90, 50/50, and 90/10. The resulting composite weights for each attraction indicator and the corresponding rankings are summarized in Tables S10 and S11. Our analysis reveals that the overall attraction rankings are relatively stable across different weight combinations, particularly for the top-ranked sites. This consistency suggests that the model's prioritization is not overly sensitive to changes in the relative emphasis of expert judgment *versus* data-driven dispersion. For example, attractions such as X and Y consistently appear within the top three across all configurations, indicating their robustness under both subjective and objective criteria. Minor variations are observed in the middle- and lower-ranked items, which is expected given the closer composite scores in those tiers. These results support the structural validity of our dual-weighting scheme and affirm that the ranking outputs are not artifacts of arbitrary parameter settings, but rather reflective of genuine multidimensional performance differences.

## Evaluation of RSR attractions

The integrated score index and ranking calculated by the TOPSIS method are the basis of the RSR algorithm. The probit value is calculated the routear regression equation is obtained by the steps, and the WRSR fitted value is obtained in accordance with the equation. The results of the RSR method and the fitted linear regression equations are shown in Fig. 3 below.

The fitted routear regression equation is as follows:

$$WRSR = -0.00934 + 0.0205 * Probit.$$

As seen from the Origin fitting, the $R^2 = 0.82095$ and the residual sum of squares is 0.00101, which indicates that the model is reasonable and reliable, with good simulation effect, and meets the needs of the evaluation model.

After building the TOPSIS-RSR model evaluation, the reasonable WRSR fitting value ranking of the attractions is obtained, based on which the attractions are graded as recommended attractions, not recommended attractions.

## Dijkstra's algorithm optimal route solution based on weight optimization

Attraction evaluation is different from route planning. Therefore, we further narrow down the focus of the six indicators, taking distance, comfort, and congestion as three key indicators for route planning, of which the comfort of the route is closely related to the travel mileage, transportation, congestion (*Shao et al., 2022*). Usually, the longer the journey time, the passenger's fatigue may increase, resulting in a reduction in the level of

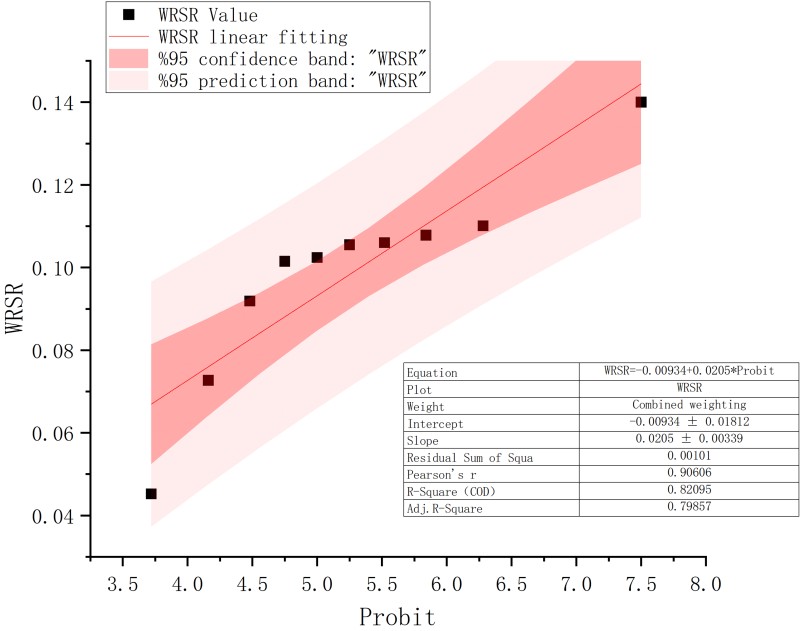

**Figure 3  WRSR fitting results.**  

comfort. Consequently, the impact of comfort can be distributed across the dimensions of distance, transportation, and congestion (*Ha, Lee & Ko, 2020*). Among these, distance and congestion emerge as the two pivotal indicators (*Yin et al., 2020*).

We divided the congestion as follows: 5 (High), 4 (Moderately High), 3 (Medium), 2 (Medium-Low), 1 (Low). Data on the distance and congestion between the eight recommended attractions has been collected. In this analysis, the weights assigned were 0.1852 for distance and 0.1468 for congestion. The path scoring table can be obtained and make the undirected graph shown in Fig. S4.

Since this undirected graph follows the 2.4 GGA step to find the optimal line:

**A -> B -> C-> G -> D -> E -> H -> F**

Namely: The Potala Palace-> Namtso Lake-> The Jokhang Temple-> Norbu Lingka-> Yamdrok Yumtso-> Bartsuncuo-> Yarlung Zangbo Daxiagu-> Mount Qomolangma

In this example study, we compare the performance of GGA and Classical Genetic Algorithm (CGA) in determining the optimal travel route. The experimental results are remarkable, as the comparison of the iteration diagrams (Fig. 4) shows that the GGA reaches the optimal solution after 533 iterations. In comparison, the CGA needs 9,671 iterations to reach the same route convergence result. Compared with the classical genetic algorithm, which required 1,872.2 iterations to converge to an optimal route solution, the proposed GGA model achieves convergence in only 103.1 iterations, resulting in a 94.45% reduction in convergence steps. This significant improvement implies not only faster computational efficiency but also the model's ability to more rapidly identify high-quality solutions that align with human satisfaction scores. In practical terms, the optimized routes generated by the GGA yield higher alignment with tourist preferences (as validated *via* expert satisfaction scoring) and reduce total transit time by avoiding redundant loops

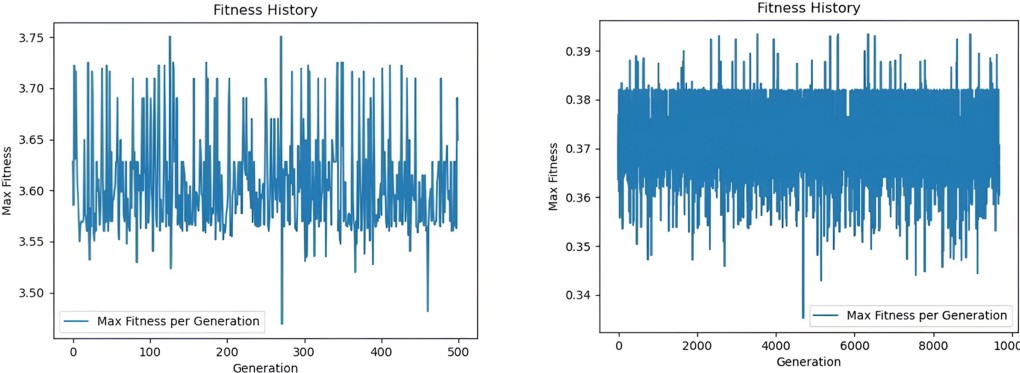

**Figure 4  Maximum fitness statistics under GGA and CGA.**

or inefficient sequences. Therefore, the improvement translates to both enhanced satisfaction and more efficient route execution in real-world scenarios.

## Modeling evaluation

Since the model is designed to create a dedicated tourist bus route based on passenger satisfaction, we analyze the results by comparing the scores of the routes generated by the model with existing popular routes in Tibet. Additionally, we conduct evaluation surveys with mainstream tourists to assess their satisfaction levels regarding the tourist route. This allows us to analyze the model's results from both perspectives.

First, we look for 30 tourists, each with corresponding traveling days and vote for the following five mainstream routes and six routes obtained from the model. The routes and statistical results are as follows (D is the model route):

- **Method 1:**

    **A:** Lhasa–Linzhi–Yarlung Zangbo Daxiagu–Namcha Barwa Peak–Lulang Forest–Bomi–Ranwu Lake in Yunnan–Laiku Glacier;

    **B:** Lhasa–Shigatse–Everest Base Camp–Jomolang Monastery–Saga–Tarkin–Gonpozi–Marbunyongtso–Layangtso–Pulan–Zada Tulin–Gugu Dynasty Ruins–Tarkin–Saga–Shigatse;

    **C:** Lhasa–Namtso–Sacred Gate of Heaven–Nagchu–For example–Sapo Mountain–For example–Nagchu;

    **D:** The Potala Palace–Namtso Lake–The Jokhang Temple–Norbu Lingka–Yamdrok Yumtso–Bartsuncuo–Yarlung Zangbo Daxiagu–Mount Qomolangma;

    **E:** Lhasa–Linzhi–Bomi–Ranwu Lake–Laiku Glacier–Lulang Forest–Yarlung Tsangpo River Grand Canyon–Namcha Barwa Peak–Sosong Village–Linzhi;

    **F:** Lhasa–Yamdrok Yumtso–Karola Glacier–Manla Reservoir–Gyangzigu Fortress–Shigatse–Zashilumbu Monastery–Tingri–Jomol Monastery–Everest Base Camp–Tingri–Saga–Tarkin–Gonchabozi– Marbunyongtso–Layangtso–Pran–Zadazhaputra–ruins of Gugu Dynasty–Tarkin–Saga–Rikaze.

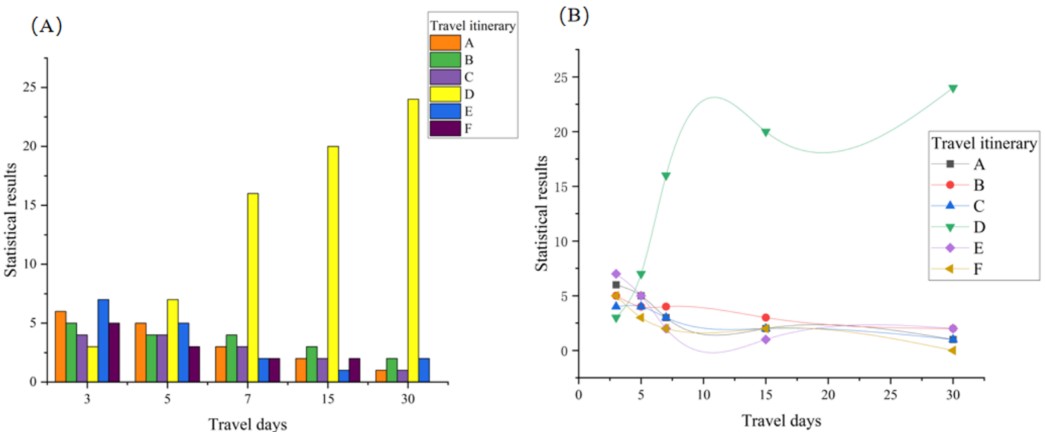

**Figure 5 Route satisfaction comparison.** (A) Shows the selection results of different routes for different tourist groups. The total statistical results are compared with the existing mainstream routes, our route result reaches 46.67% regarding the comprehensive tourist selection rate, far exceeding the other five major routes. In addition, we also further intuitively show the comparison results of the advantages and disadvantages of the line through the curve in (B). Through randomly conducted surveys of the public and experts, the line satisfaction is as high as 94.6%, the combination of which fully validates the scientific and rational nature of our method.

The statistics and comparison results are presented in Figs. 5, including the number of travel days for tourists other than the typical 3-day trips. The D route received the fewest votes overall, while other categories of tourists on the D route received the highest number of votes. The D route garnered 70 votes, accounting for 46.67% of the overall votes, significantly surpassing the other five routes.

- **Method 2:**

We conducted route evaluation for mainstream tourists (traveling time of 7 days). After transmitting 330 questionnaires and 20 expert consultations, the total number of people who were satisfied with the route was 331, with a satisfaction rate as high as 94.6%.

The results obtained from the above two surveys comprehensively verified that the route obtained from the model has a high level of tourist satisfaction, proving the scientificity and accuracy of the route evaluation method.

## DISCUSSION

In this study, we first compared the performance of multiple machine learning algorithms for review sentiment analysis. After careful evaluation, we found that the SVM model performs the best. Therefore, we chose SVM as the base model for the subsequent analysis, through which we adopted the subjective-objective comprehensive assignment method combining AHP subjective assignment method and entropy weighting method after identifying six key indicators such as tickets, accommodation, preference, motivation, distance traveled and crowding. This method combines the expertise and subjective insights of industry professionals with the objective characteristics of the data. It ensures a

fair distribution of weights for each influential indicator, thereby enhancing the accuracy and fairness of the evaluation results.

Based on the determination of the comprehensive weights, we constructed the TOPSIS-RSR combination evaluation model. This model combines the advantages of the TOPSIS method in calculating similarity and ranking and the simplicity of the RSR method in auxiliary grading, effectively overcoming the inability to determine the reasonableness of the evaluation results of TOPSIS and its shortcomings in grading. Through this method, we can obtain precise grading results of attractions, improving the evaluation process's accuracy and rationality.

After identifying the attractions, we analyzed the key factors that affect tourist route satisfaction. Through comprehensive analysis and screening, we determined that distance traveled and crowding are the two primary indicators influencing route satisfaction. We first improved the classical greedy algorithm using comprehensive weights and then introduced the idea of a genetic algorithm to further optimize the greedy algorithm. The GGA improves the convergence speed by 94.489% over the classical algorithm, effectively avoiding the local optimal solution problem that the classic greedy algorithm may fall into. Compared with the existing mainstream routes, our route result reaches 46.67% regarding the comprehensive tourist selection rate, far exceeding the other five major routes. In addition, through randomly conducted surveys of the public and experts, the line satisfaction is as high as 94.6%, the combination of which fully validates the scientific and rational nature of our method. To improve statistical transparency, we also report variability measures: the 46.67% route selection rate had a standard deviation of 5.81% and a 95% confidence interval of [39.42%, 53.92%]. The 94.6% satisfaction score corresponded to a standard deviation of 3.12% and a 95% confidence interval of [91.58%, 97.62%].

However, the model also has shortcomings. If it is to be generalized to different regions for tourism route planning, a lot of local fieldwork is needed to consider more comprehensive factors. The model is hoped to be improved and refined in subsequent studies to provide more thinking directions for scientific and reasonable tourism route planning. While the proposed model demonstrates strong empirical performance—achieving a 46.67% route selection rate and a 94.6% overall satisfaction rate—there are limitations that merit discussion. First, the tourist and expert survey sample sizes (30 individuals for route voting and 150 for indicator evaluation) are relatively modest, which may limit the statistical generalizability of our conclusions. Although satisfaction rates were high, we acknowledge that larger sample sizes across multiple demographics would be necessary to further validate these findings. Additionally, variability in satisfaction responses was not explicitly measured *via* standard deviation in this study. While the binary outcome (selected *vs.* not selected, satisfied *vs.* unsatisfied) provides useful trends, future work could incorporate Likert-scale ratings to capture variance in subjective experience more precisely. Lastly, although the model framework is designed for generalizability, its current validation is geographically constrained to the Tibet region. We therefore temper our claims about broad adaptability and note that real-world deployment in other regions would require localized retraining and validation using context-specific data.

## CONCLUSIONS

This study presents an innovative approach to optimizing tourist bus route planning by integrating machine learning algorithms, multi-criteria evaluation methods, and an enhanced GGA. The proposed model effectively combines subjective and objective indicators to enhance the tourist experience while promoting sustainable tourism practices. The results highlight the superiority of the SVM model for sentiment analysis, as it identifies key satisfaction factors used in the TOPSIS-RSR framework to evaluate and prioritize attractions. With its improved convergence speed, the GGA offers a practical solution for optimizing routes that maximize tourist satisfaction, achieving a 94.6% satisfaction rate.

Furthermore, while the model structure shows theoretical adaptability, its current validation is geographically confined to the Tibet region. As such, we acknowledge that practical deployment in other regions would require localized data collection, customization of indicator sets, and empirical re-validation to account for regional tourism preferences, infrastructure, and cultural contexts. Therefore, broader applicability remains a promising direction for future investigation rather than an established claim. While the model has shown considerable success in optimizing tourist routes, future research should explore its application in different geographical and cultural settings, incorporating additional variables to refine its accuracy and efficiency. This work contributes to the growing body of research on sustainable tourism planning by offering a robust, data-driven solution for creating optimized, environmentally conscious travel routes that enhance both visitor satisfaction and the longevity of tourist destinations. Note that the current validation is limited to the Tibet Autonomous Region. While the proposed framework is theoretically adaptable, local replication with region-specific indicators and tourist feedback is essential before generalizing to other areas. Future extensions of the model may consider integrating qualitative experience factors such as scenic value or visual richness of routes. Since tourist satisfaction is not solely determined by travel speed or itinerary efficiency, incorporating experiential dimensions could further improve the model's alignment with real-world traveler preferences.

### Funding

This research was funded by Research on the Optimisation of Self-guided Tour Routes Based on the Tibet Tourism Traffic Network Diagram, Natural Science Foundation of the Tibet Autonomous Region, China, grant number XZ202201ZR0038G. The funders had no role in study design, data collection and analysis, decision to publish, or preparation of the manuscript.

### Grant Disclosures

The following grant information was disclosed by the authors:
Research on the Optimisation of Self-guided Tour Routes Based on the Tibet Tourism

Traffic Network Diagram, Natural Science Foundation of the Tibet Autonomous Region, China: XZ202201ZR0038G.

## Competing Interests

The authors declare that they have no competing interests.

## Author Contributions

- Suping Cui conceived and designed the experiments, prepared figures and/or tables, and approved the final draft.
- Xiang Zhang conceived and designed the experiments, performed the experiments, prepared figures and/or tables, and approved the final draft.
- Haiqiong Liang performed the experiments, analyzed the data, authored or reviewed drafts of the article, and approved the final draft.
- Chang Liu analyzed the data, performed the computation work, authored or reviewed drafts of the article, and approved the final draft.
- Sa Du analyzed the data, performed the computation work, prepared figures and/or tables, and approved the final draft.
- Boyu Hou performed the computation work, authored or reviewed drafts of the article, and approved the final draft.
- Xinyan Wang conceived and designed the experiments, authored or reviewed drafts of the article, and approved the final draft.
- Zhongfeng Wu performed the computation work, prepared figures and/or tables, authored or reviewed drafts of the article, and approved the final draft.

## Ethics Statement

All questionnaire surveys and expert interviews conducted in this study were performed in accordance with relevant ethical guidelines and regulations. Ethical approval was obtained from the Ethics Committee of Tibet University. All participants were recruited voluntarily, informed of the purpose of the study, and provided written informed consent before participation. All collected data were anonymized and used exclusively for academic research purposes.

## Data Availability

The code is available at GitHub and Zenodo:

- https://github.com/zxds1234/Code_pjcs.git.

. zxds1234. (2025). zxds1234/Code_pjcs: GGA-DOI (v1.0.0). Zenodo. https://doi.org/10.5281/zenodo.15728560.

The raw data are available in the Supplemental File.

## Supplemental Information

Supplemental information for this article can be found online at http://dx.doi.org/10.7717/peerj-cs.3221#supplemental-information.

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
