# Peer review of "A planning model for dedicated tourist bus routes based on an improved genetic-greedy algorithm and machine learning"

_PeerJ Computer Science, doi:10.7717/peerj-cs.3221_

## Round 0.1 · original submission · Major Revisions

**Language Note:** The review process has identified that the English language must be improved. PeerJ can provide language editing services - please contact us at [email protected] for pricing (be sure to provide your manuscript number and title). Alternatively, you should make your own arrangements to improve the language quality and provide details in your response letter. – PeerJ Staff

Reviewer 1 ·

Basic reporting

The manuscript presents an ambitious model combining machine learning, multi-criteria decision-making (MCDM), and route optimization techniques to improve tourist bus route planning. While the core idea is relevant and innovative, there are several areas where the manuscript would benefit from substantial improvement to meet scholarly standards of clarity, rigor, and transparency.

Firstly, the English language usage requires refinement. The manuscript currently contains numerous monotonous sentence structures and unnecessary filler phrases such as "in summary" that reduce the work's overall readability and professional tone. Greater variation in sentence construction and a more concise style would help improve clarity and engagement. I recommend that the authors seek language editing support through a professional service or by a native speaker familiar with technical writing in computer science and engineering.

Regarding the structure of the introduction, while the authors appropriately emphasize the importance of user satisfaction in the context of sustainable tourism, the section fails to clearly articulate the research question. A well-defined research question is fundamental for contextualizing the methodological choices and assessing the study’s contributions. The introduction should explicitly state what specific problem the study addresses, how existing approaches fall short, and what this paper proposes to do differently. Framing this initially will help readers understand the proposed model's rationale and novelty.

A major shortcoming of the methodology section is the lack of justification for using AHP (Analytic Hierarchy Process) and the Entropy Weight Method. These powerful tools within MCDM are not universally appropriate for all decision contexts. The authors should provide a rationale for selecting AHP, explaining its relevance and advantages over other MCDM approaches. Additionally, it is important to address the limitations of the research design in comparison to other alternatives.

Experimental design

Regarding methodological rigor, the manuscript incorporates multiple techniques, including sentiment analysis using machine learning, a hybrid AHP and entropy weighting system, and a modified genetic-greedy algorithm (GGA). However, several aspects of the investigation fall short of the standards required for reproducibility and technical transparency.

First, the rationale for choosing AHP and the entropy weight method is not sufficiently discussed. These methods have theoretical assumptions and practical limitations that should be acknowledged. The manuscript should explain why these were chosen over other multi-criteria decision-making approaches and should provide evidence of their suitability for the given problem. For improvement, the authors may refer to relevant methodological literature, such as Munier and Hontoria (2021) on AHP and Zhu et al. (2020) on entropy weighting, to provide a foundation for their methodological choices.

Additionally, the machine learning section lacks critical implementation details. While SVM is identified as the best-performing classifier, the manuscript does not specify the parameter settings used (e.g., kernel type, regularization parameters), the evaluation metrics beyond R², or the structure of the AdaBoost model. There is also ambiguity around how features were selected and how sentiment was operationalized into satisfaction indicators. These omissions hinder the replicability of the approach and limit its technical credibility.

Another important concern is the absence of an ethics statement. The study involves data collected from human subjects via questionnaires and expert evaluations. However, the manuscript does not indicate whether ethical approval was obtained, how participants were recruited, or whether informed consent was acquired. This represents a significant oversight, especially given the increasing emphasis on ethical transparency in computational research involving human data.

To enhance reproducibility and integrity, the authors should:

Clearly state the research question and knowledge gap.

Justify the selection of AHP and entropy weight methods using relevant literature.

Provide comprehensive methodological details for all machine learning components.

Describe the design, execution, and validation of the genetic-greedy algorithm in replicable terms.

Include a formal ethics statement detailing consent procedures and approval bodies.

Without these improvements, the study does not meet the required technical and ethical rigor standards expected in scholarly publications.

Validity of the findings

The manuscript presents performance evaluations for several machine learning models, ultimately selecting SVM based on classification accuracy for sentiment analysis. However, the statistical results reported are problematic. Specifically, the R-squared (R²) values provided for the models are extremely low, approaching zero, which indicates a weak fit and undermines confidence in the reliability of the sentiment classification component. The authors must address this issue in the results section, explaining the implications of such low explanatory power and clarifying how model selection is justified despite these values.

I understand that the r values correspond with sentiment analysis, and then they apply classification models. However, all these flaws must at least be discussed. Also, mention AdaBoost, but do not elaborate on it being an ensemble machine learning technique based on others, such as decision trees.

Additional performance metrics, such as F1-score, precision, recall, or confusion matrices, should be reported to provide a more comprehensive assessment of classification effectiveness.

Moreover, the manuscript lacks a critical discussion of the uncertainty and variability associated with its multi-criteria evaluation framework. For example, while the study combines subjective (AHP) and objective (entropy) weights to rank attractions, it does not include sensitivity analysis or error estimation to demonstrate the stability of the rankings under different weight configurations. This omission makes it difficult to assess the robustness of the final route recommendations.

The survey data used to validate the model’s satisfaction rates (46.67% selection rate and 94.6% satisfaction rate) are compelling but not contextualized with details on response variability or standard deviations. Additionally, the modest sample size (e.g., 30 tourists per route evaluation, 150 total questionnaire responses) limits the generalizability of the findings. These limitations should be clearly acknowledged in the discussion, and claims about the model’s adaptability to other regions should be tempered accordingly.

The conclusions generally align with the study's goals and are summarized effectively. However, they occasionally extend beyond what is directly supported by the data. For example, while the case study demonstrates promise, broader claims about global applicability or model generalizability are premature without further empirical validation in diverse geographic and cultural settings.

To strengthen the validity of the findings, I kindly encourage the authors to :

- Address the low R² values and supplement with additional model evaluation metrics.

- Include sensitivity analysis or robustness checks for the MCDM framework.

- Discuss the limitations imposed by the sample size and the geographic specificity of the data.

- Ensure the presented results clearly support all conclusions and refrain from overgeneralizing.

By refining the statistical presentation and tempering the generalizations, the authors can better align their conclusions with the evidence provided and strengthen the study’s scientific reliability.

Additional comments

The authors are advised to carefully re-examine all figures in the manuscript to ensure visual consistency, clarity, and completeness. For example, Figure 1 is currently missing a connection line between two key steps, which disrupts the logical flow and makes the process difficult to interpret. Such omissions can compromise the reader's understanding of the proposed algorithm and must be corrected. Another example lies in Figure 2, which comprises two subplots but lacks individual titles or labels for each subplot. This omission diminishes the clarity of the comparative analysis and may lead to confusion about what each part of the figure represents. Each subplot should be clearly labeled, and the overall figure caption should include a brief explanation of the subplots to enhance interpretability.

Beyond visual presentation, the manuscript is also missing an essential ethical component. The research involves collecting and analyzing responses from human participants through surveys and expert evaluations. However, there is no mention of ethical approval, informed consent procedures, or compliance with institutional review board (IRB) standards. This is a critical oversight. Ethical approval and transparency regarding participant consent are non-negotiable aspects of research involving human data and must be explicitly addressed in the methods section. The authors should indicate whether ethics approval was sought, from which body, and how participants obtained consent.

To improve the methodological rigor and theoretical grounding of the manuscript, the authors are encouraged to consult the following literature:

Munier, N., & Hontoria, E. (2021). Uses and Limitations of the AHP Method. Springer.
This book comprehensively explores the theoretical boundaries and practical applications of the AHP methodology, highlighting where it is most effective and where caution is warranted.

Zhu, Y., Tian, D., & Yan, F. (2020). Effectiveness of entropy weight method in decision-making. Mathematical Problems in Engineering, 2020, 1–5. https://doi.org/10.1155/2020/3564835
This article critically evaluates the advantages and limitations of the entropy weight method and is particularly useful for understanding its suitability and potential flaws in decision-making frameworks.

Talero-Sarmiento, L., Gonzalez-Capdevila, M., Granollers, A., Lamos-Diaz, H., & Pistili-Rodrigues, K. (2024). Towards a Refined Heuristic Evaluation: Incorporating Hierarchical Analysis for Weighted Usability Assessment. Big Data and Cognitive Computing, 8(6), 69. https://doi.org/10.3390/bdcc8060069

This work illustrates an applied use of multi-criteria decision-making (MCDM) incorporating AHP, and offers a robust example of how to integrate and justify MCDM methodologies in applied research.

·

Basic reporting

Basically, the language for the manuscript needs improvement, especially in the abstract and introduction sections. Please avoid using we; it's better to use a different style of writing, such as using the third person in the manuscript.

Overall, the literature is sufficient for the manuscript. However, it would be better to highlight why bus planning routes for tourists is very important so that the authors need to provide a new algorithm just to solve the problem. Is it due to congestion?

overall the purpose algorithm is not new but its application is new for the bus route

overall the figures ;provided in the manuscript is good quality execept figure 1

Experimental design

From my point of view, this manuscript is inline with the journal's aims and scope.

Research question and objective is well design full stated to find solution for the research gap however, it is best if the authors could provide the problem statement clearly in the manuscript

The authors trying to solve the issue with an improvement of the Greedy genetic algorithm whereby they incorporated weight of alpha and betha for the solution as stated in equation 1.

Validity of the findings

The verification of the model was compared to the classical genetic algorithm 1.8722 with is 94.448% improvement. I think it is better to state 94.45% improvement from which number. How significant is 94.45% to the bus routes? the improvement in terms of human satisfaction or in terms of time travel?
Sometimes the satisfaction of the tourist is not totally reliant on how fast the bus is moving; maybe the scenery would be one of the items that should be included in the study.

The current conclusion and discussion were reflected to the question, problem and supported by the results. However there is several question that the authors need to answer regarding the results from the study.

---

## Round 0.2 · Minor Revisions

Please address the remaining points from the reviewer.

Reviewer 1 ·

Basic reporting

The manuscript is now written in generally clear, technically sound English, and its overall tone meets professional standards.

A final light copy-edit—mainly to trim a few lingering redundancies such as recurrent transition phrases and to harmonise tense use—would lift the prose to an excellent level of polish.

The introduction provides a cogent rationale for the study and demonstrates familiarity with the broader tourism-routing literature. Recent critical sources on AHP and entropy weighting have been incorporated, ensuring a sufficient and appropriately referenced background context. Structurally, the paper follows the conventional sequence of Introduction, Methods, Results, Discussion, and Conclusion, making it easy to navigate.

Most visual materials have been improved, yet two small presentation details remain: the y-axis label in the attraction-preference bar chart still reads “Voute Count” instead of “Vote Count”, and the two supplementary pie charts would benefit from explicit titles and a higher-contrast (or patterned) colour palette to make adjacent slices immediately distinguishable, particularly for colour-blind readers.

The raw data and code are now deposited on Zenodo and linked from the text, thereby satisfying PeerJ’s open-data requirements.

Taken together, the submission is self-contained, presents all results germane to the stated hypothesis, and does not appear to fragment a larger study. Once these minor textual and figure refinements are completed, the manuscript will meet the journal’s basic reporting standard in full.

Experimental design

The revised manuscript now articulates a focused research question: how to embed dynamic tourist-satisfaction signals into group-tour bus-route optimization, which is both relevant to the journal’s scope and positioned as a response to the shortcomings of existing static or cost-only routing models. The methodological framework—combining sentiment-driven indicator extraction, an AHP–entropy weighting scheme, and an improved genetic-greedy algorithm—demonstrates technical rigour and is, in principle, reproducible. Implementation details for the machine-learning components, weighting calculations, and optimisation steps are now described with sufficient specificity, and the accompanying code repository further supports replicability. Ethical compliance has also been addressed through the addition of an institutional approval statement and a description of informed consent.

Two clarifications would strengthen the experimental design: first, please outline how the 150 questionnaire respondents were recruited (for example, on-site convenience sampling versus online outreach) so readers can gauge potential sampling bias; second, indicate which survey items were quantitatively integrated into the model pipeline—such as whether travel-duration or attraction-scope answers influenced feature weighting or merely served descriptive purposes.

With these brief additions, the study will meet PeerJ’s standards for methodological transparency and ethical robustness.

Validity of the findings

The evidence presented now supports the central claim that a sentiment-informed, hybrid-weighted optimisation framework can generate tourist-bus routes preferred by surveyed travellers. The underlying data and code are openly archived, and the switch to classification-appropriate metrics (precision, recall, and F1) makes the sentiment-model comparison statistically defensible. The sensitivity test of the AHP–entropy weights further demonstrates that attraction rankings are not artefacts of a single parameter choice.

Nevertheless, a few refinements would increase confidence in the robustness of the findings. First, because the route-voting experiment involved only 30 tourists and the indicator survey 150 respondents, please provide basic dispersion measures—such as standard deviation or 95% confidence intervals—for the reported 46.67% route-selection rate and 94.6% satisfaction figure. These variability indicators will let readers judge whether the high percentages are stable or might fluctuate in a larger sample. Second, clarify how any outliers or inconsistent responses were handled during data cleaning; this is particularly relevant for open-ended questionnaire items.

Finally, while the conclusions rightly avoid sweeping generalisation, they could acknowledge more explicitly that the current validation remains geographically specific to Tibet and that replication in other regions will be necessary before broader claims can be made.

With these modest statistical disclosures and a slightly sharper statement of scope, the study’s conclusions will be fully aligned with the evidence presented.

Additional comments

The manuscript has advanced substantially since the first round, and I commend the authors for the care taken to clarify the research question, justify the methodological choices, and add the required ethical and reproducibility information. In its current form, the work is nearing publication.

Only a handful of presentation issues remain—chiefly the small figure corrections (the “Vote Count” label, clearer titles and contrast for the supplementary pie charts, and a less crowded network diagram) — and the inclusion of basic variability statistics for the survey-based results.

Addressing these points will not alter the substance of the contribution but will ensure the paper fully meets the journal’s standards for clarity and transparency.

With these minor revisions incorporated, I believe the article will be ready for acceptance.

---

## Round 0.3 · accepted · Accept

Dear Authors,

Thank you for clearly addressing the reviewers' comments. Your paper now seems sufficiently improved and ready for publication.

Best wishes,

Reviewer 1 ·

Basic reporting

There are no further improvements needed.

Experimental design

-

Validity of the findings

-